# Thermomechanical Fatigue of Lost Foam Cast Al–Si Cylinder Heads—Assessment of Crack Origin Based on the Evaluation of Pore Distribution

**Martin Wagner [1],\*, Andreas Mösenbacher [1] , Marion Eiber [1], Martin Hoyer [2] , Marco Riva [2] and Hans-Jürgen Christ [3]**

1   Method Development & Fatigue Strength, IABG mbH, 85521 Ottobrunn, Germany
2   Simulation Structural Mechanics Powertrain, BMW AG, 80788 München, Germany
3   Institut für Werkstofftechnik, Universität Siegen, 57076 Siegen, Germany
\*   Correspondence: Wagnerm@iabg.de; Tel.: +49-89-6088-2847

**Abstract:** In automotive cylinder heads, thermomechanical fatigue (TMF) leads to crack initiation within the critical loaded sections. This effect becomes even more relevant in lost foam cast cylinder heads since its system-dependent porosity shows a significant influence on the lifetime under TMF loading. This work covers the identification of a criterion for crack initiation in order to provide the basis for an effective quality control with improved statistical safety by nondestructive testing. Specimens extracted from lost foam cylinder heads were investigated by uniaxial TMF tests, X-ray micro computer tomography (μCT), and scanning electron microscopy (SEM). Due to pore analyses on a global and local scale, it is concluded that pore networks are crucial for crack initiation. Thus, a tool for computation of pore accumulations from μCT data containing interaction criteria by Murakami was developed in order to assess the crack origin. The consideration of pore accumulations significantly improves the predictive accuracy compared to the consideration of single pores.

**Keywords:** aluminum-silicon cylinder head; lost foam; pore accumulation; pore distribution; thermomechanical fatigue; X-ray micro computer tomography

## 1. Introduction

In automotive engine components such as cylinder heads, aluminum alloys are widely spread due to their good mechanical and physical properties, along with suitable manufacturing characteristics and the excellent suitability for lightweight design [1]. In cylinder heads, temperature cycles (engine start-stop, full load, partial load) in combination with high temperature gradients lead to thermomechanical fatigue (TMF). The result is crack initiation within the critical section on the combustion chamber side [2] and integrated exhaust manifolds. In the past, the thermomechanical fatigue behavior of cylinder heads was investigated in detail [2,3].

Cylinder heads are increasingly manufactured by lost foam rather than gravity die casting due to process cost reduction, geometry optimization, and consumption control [4,5]. However, the result of systemic lower cooling rates is a coarser microstructure as well as an increase in number, size, and volume fraction of pores [6,7]. Besides rather spherical gas pores, the solidification of the melt causes extensive shrinkage pores with small sphericities [7] associated with high stress intensities. A few studies [7,8] deal with the 3D in situ analysis of the crack initiation and growth caused by shrinkage pores in lost foam cast components. The TMF life of lost foam cast cylinder heads can be assessed by using energy based damage models with additional consideration of the local damage [5]. There is a significant variation of the lifetime of identically loaded sections on the combustion chamber side of

cylinder heads. It is well studied that the interaction of defects plays an important role with regard to crack initiation [9–11]. Due to the presence of large pore numbers and extensive shrinkage pores, the interaction might be the main reason for lifetime variation.

It is common to use statistical methods to evaluate pore distributions because of their suitability for volume extrapolation [12,13]. Romano et al. [14] worked on the statistical evaluation of single defects within additive manufactured parts using exponential and extreme value distributions. Furthermore, extreme value models for estimating defect size in clean steels were developed for multiple types of large inclusions [15]. In the past, many authors studied the influence of material defects on the LCF lifetime and the fatigue limit in the HCF regime. Charkaluk et al. [16] determined a lifetime probability density function from a probability density function of defects using a growth law. Rödling et al. [17] investigated the influence of critical non-metallic inclusions in high-strength steels on HCF design properties by means of statistics of extremes. Regarding an aluminum alloy, doubling the initial defect diameter causes an approximate 30% decrease in run-out stress [18]. These studies [17,18] underlie the recognition that, when defects become the fracture origin, crack initiation occurs at the largest defect or the critical combination of defects within the volume [9].

However, there exists a strong need to systematically and effectively qualify large sample sizes of TMF-loaded lost foam cast components for industrial applications, by means of nondestructive testing instead of time-consuming and cost-intensive destructive testing. The application of X-ray micro computer tomography (μCT) enables the nondestructive detection of defects within the volume of components, even inside inaccessible sections. The detection of defects by μCT requires a significant difference in density to the surrounding material and is limited by the resolution of the μCT system. In this study, the application of μCT is well suited as the porosities in lost foam cast components have many times larger expansions than the resolution of the used μCT system (10 μm).

The aim of this work is the identification of a criterion for crack initiation based on μCT defect data. Due to the presence of large pore networks caused by the lost foam casting process, defect interaction criteria by Murakami [9] are used. Subsequently, a correlation between the criterion for crack initiation and the lifetime has to be derived in order to assess the lifetime variation of cylinder heads as a function of the variation of the criterion in specimen geometry.

An extensive experimental program comprising TMF testing, scanning electron microscopy (SEM) of the fracture surface, and μCT scans before and after TMF testing is conducted. On a global scale, the pore volume fraction and number of pores are analyzed. On a local scale, the characteristic parameters of single pores are statistically evaluated by means of probability distributions. A tool for the computation of pore accumulations is developed in order to assess the crack origin.

## 2. Materials and Methods

### 2.1. Material

Lost foam cast cylinder heads were taken from the series process to consider the actual present casting conditions and the resulting quality of the material. The cylinder heads are manufactured from the precipitation-hardening aluminum-silicon alloy AlSi7MgCu0.5 with heat treatment T5. The specimens were extracted from the region between the intake and exhaust port of the cylinder heads (Figure 1) to consider the influence of the lost foam casting process on the lifetime under TMF loading. Thus, the specimens have small dimensions. The total specimen length is 80 mm, while the mount and test diameter read 10 mm and 7 mm, respectively.

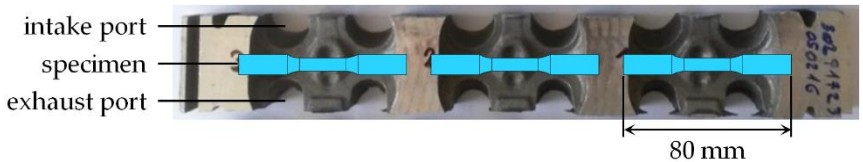

**Figure 1.** Specimen extraction.

### 2.2. TMF Tests

Strain controlled uniaxial TMF tests were performed with a servo-hydraulic test rig (MTS Systems GmbH, Berlin, Germany). For strain measurement a high-temperature extensometer with a gauge length of 12 mm was used. The strain control was realized by the software MTS 793 (software version 5.9, MTS Systems GmbH, Berlin, Germany). The heating of the specimen was conducted by a 10 kW high-frequency generator and an induction coil. In order to be able to perform TMF tests with small specimens (Section 2.1), the induction coil consists of two pipes: an outer water-cooled pipe and an inner compressed air pipe with nozzles for air transportation to the specimen. A flattened thermocouple (type K) with 0.1 mm thickness was used for temperature measurement (Figure 2). The test procedure and temperature control were realized by the software LabVIEW (software version 2014, National Instruments Germany GmbH, Munich, Germany).

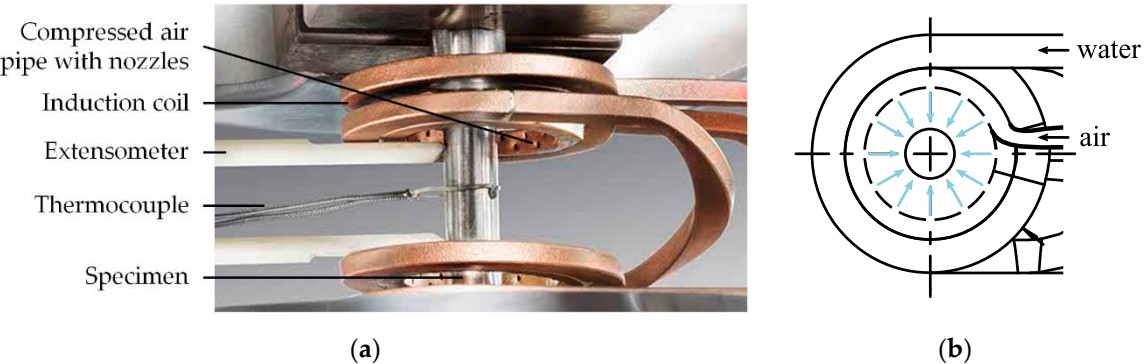

**Figure 2.** (**a**) Thermomechanical fatigue (TMF) test setup. (**b**) Induction coil with outer water-cooled pipe and inner compressed air pipe with nozzles for air transportation to the specimen.

Twenty-one TMF tests of specimens extracted from two batches were performed with a minimum temperature of 50 °C and two different maximum temperatures (225 °C and 250 °C) with heating and cooling rates of 5 °C/s using a triangle shaped signal. Three mechanical strain range values $\Delta\varepsilon_{mech,1}$, $\Delta\varepsilon_{mech,2}$, and $\Delta\varepsilon_{mech,3}$ were applied based on the engine service conditions. The strains are scaled with respect to $\Delta\varepsilon_{mech,2}$ according to $\Delta\varepsilon_{mech,1} = 1.2\,\Delta\varepsilon_{mech,2}$ and $\Delta\varepsilon_{mech,3} = 0.8\,\Delta\varepsilon_{mech,2}$. The maximum mechanical strain of each cycle was defined as $\varepsilon_{mech,max} = 0$. A hold time of 60 s was applied at the maximum temperature in order to cover a worst case automotive usage profile with regard to creep damage, i.e., during the TMF tests the specimen is exposed to the same amount of creep damage as the cylinder head under worst case service conditions. The TMF tests were carried out in out-of-phase (OP) mode, i.e., with a phase shift of 180° between the thermal and mechanical strain (Figure 3).

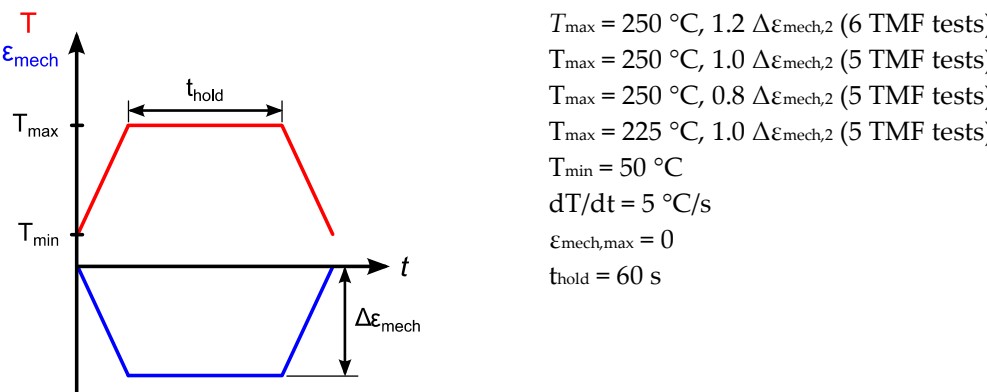

$T_{max}$ = 250 °C, 1.2 $\Delta\varepsilon_{mech,2}$ (6 TMF tests)
$T_{max}$ = 250 °C, 1.0 $\Delta\varepsilon_{mech,2}$ (5 TMF tests)
$T_{max}$ = 250 °C, 0.8 $\Delta\varepsilon_{mech,2}$ (5 TMF tests)
$T_{max}$ = 225 °C, 1.0 $\Delta\varepsilon_{mech,2}$ (5 TMF tests)
$T_{min}$ = 50 °C
$dT/dt$ = 5 °C/s
$\varepsilon_{mech,max}$ = 0
$t_{hold}$ = 60 s

**Figure 3.** Loading conditions of performed TMF tests.

### 2.3. Microstructure Analyses

After TMF testing, the fracture surfaces were investigated by SEM (Carl Zeiss AG, Oberkochen, Germany) in order to identify the crack origin of each specimen. All fracture surfaces show striations (Figure 4a), and fracture propagation lines are detected near material defects. Figure 4a shows freely solidified surfaces which occur due to shrinkage of the material during the casting process and result in shrinkage pores. The system-dependent, slow cooling rate of the lost foam casting process increases the trend to the evolution of freely solidified surfaces and thus shrinkage pores.

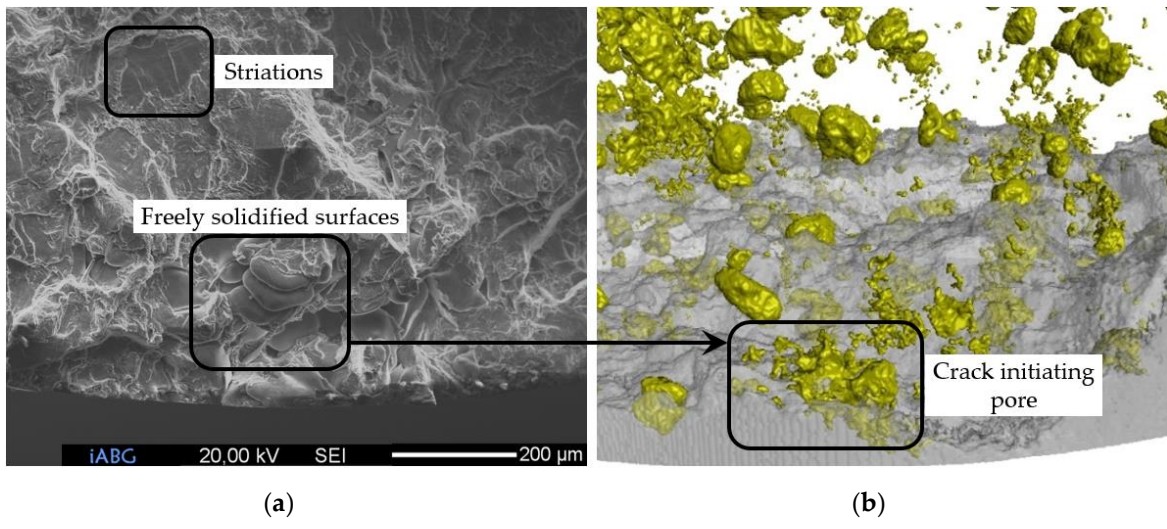

(**a**) (**b**)

**Figure 4.** (**a**) Crack origin detected by scanning electron microscopy (SEM); (**b**) Crack-initiating pore detected by X-ray micro computer tomography (μCT).

μCT scans with a resolution of 10 μm were performed by means of a X-ray inspection system (GE Sensing & Inspection Technologies GmbH, Wunstorf, Germany) before and after the TMF tests. The μCT scan before TMF testing was performed across the entire gauge length of the specimen to consider all material defects, including pores open to the specimen surface. Moreover, the pore volume fraction and the number of pores within the volume were determined. The scan after TMF testing focused on the fracture area. For μCT data analysis the software VGSTUDIO MAX (software version 2.2, Volume Graphics GmbH, Heidelberg, Germany) was used. The volumes before and after TMF testing were virtually overlapped. In order to achieve this, both volumes were manually positioned to each other by aligning the surfaces of conspicuous pores, followed by an automatic alignment of both volumes by the evaluation software. Subsequently, the fracture surfaces scanned by SEM and μCT were visually compared. The crack-initiating pores were detected in the μCT scan looking on the location of the actual crack origin, i.e., freely solidified surfaces, detected by SEM (Figure 4b) [19].

Additionally, all pores located on the fracture surface were detected by the following method: after the described virtual overlap of the volumes before and after TMF testing, some pores detected in the μCT scan before TMF testing did not touch the fracture surface, even though SEM investigations obviously showed that they are located on the fracture surface. It is assumed that this small shift between the pores and the fracture surface is caused by the plastic deformation of the material close to the fracture surface during the TMF test. In order to consider these pores the smallest distance of each pore to the fracture surface was computed. All pores closer than 50 μm to the fracture surface were defined as pores on the fracture surface. The value of 50 μm (five times the scan resolution) was visually adjusted. By means of this procedure, the influence of inner pores on the crack initiation behavior can be evaluated.

## 3. Experimental Results

### 3.1. TMF Tests

In Figure 5, the relation between the mechanical strain range (scaled) and lifetime (number of cycles to a stress drop of 20%) is shown for all TMF tests performed. There is no influence of the batch on the lifetime. However, the lifetime increases with increasing maximum temperature. There are lower maximum stresses, but a larger plastic strain range with increasing temperature. Thus, it is assumed that the maximum stresses have a greater influence on the lifetime than the plastic strain range.

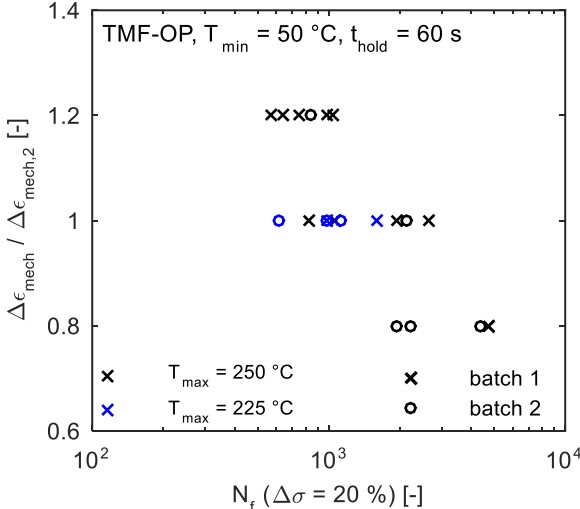

**Figure 5.** Relation between mechanical strain range (scaled) and lifetime for all TMF tests performed.

Figure 6 provides the cyclic hardening–softening curves and the stress–strain hystereses of cycle $N = 500$ for the performed TMF tests with a mechanical strain range of $\Delta\varepsilon_{mech} = 1.2\ \Delta\varepsilon_{mech,2}$ and a maximum temperature of $T_{max} = 250\ °C$. All stresses are scaled to the mean stress range $\Delta\sigma_2$ at half number of cycles to failure for the TMF tests with a mechanical strain range of $\Delta\varepsilon_{mech,2}$. Each TMF test shows a positive mean stress due to out-of-phase loading. The nominal maximum and minimum stresses of each cycle decrease with increasing cycle number due to ageing effects. The nominal maximum and minimum stresses lie within a scatter band of 10% and the lifetime varies from about 500 to 1000 cycles (Figure 6a). Additionally, the shape and area of the stress–strain hystereses as well as the stress relaxation during the hold time are almost identical (Figure 6b). Thus, there is no appreciable influence of the nominal maximum and minimum stresses as well as the characteristics of the stress–strain hystereses on the lifetime.

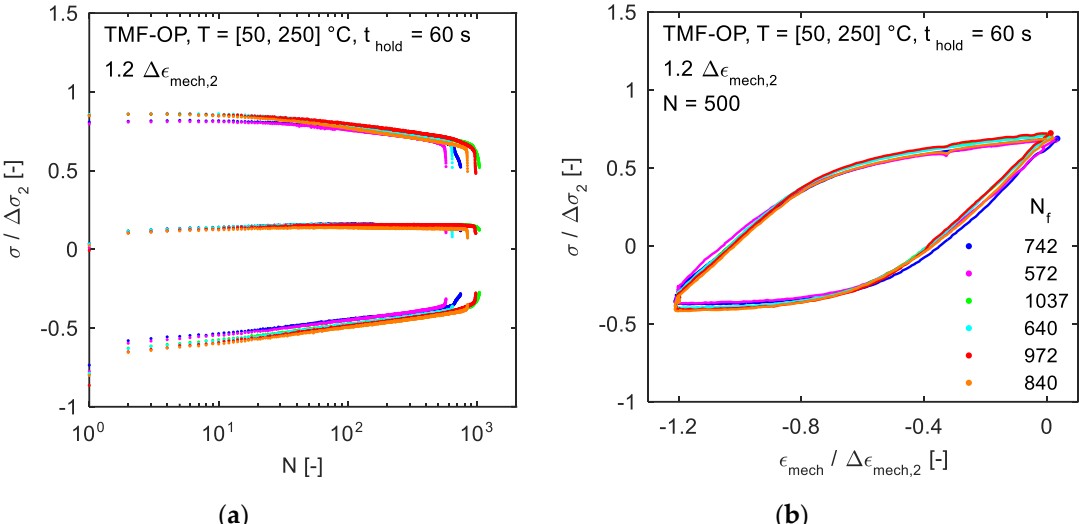

**Figure 6.** Performed TMF tests with $\Delta\varepsilon_{mech} = 1.2\ \Delta\varepsilon_{mech,2}$ and $T_{max} = 250$ °C: (**a**) Cyclic hardening-softening curves; (**b**) Stress–strain hystereses of cycle N = 500.

### 3.2. Microstructure Results

SEM investigations of the fracture surface showed that, in all specimens, cracks initiate from macro-scale porosities. Thus, there is no influence of the applied mechanical strain range ($\Delta\varepsilon_{mech,1}$, $\Delta\varepsilon_{mech,2}$ and $\Delta\varepsilon_{mech,3}$) and the maximum temperature (225 °C and 250 °C) on the type of crack initiation. Except for a few specimens where cracks initiate from an extensive single pore (Figure 4), mostly a pore network consisting of several pores was identified at the crack origin. It is assumed that adjacent pores coalesce during a few cycles and therefore act as crack initiator. In most specimens, pore networks consist of shrinkage pores, but also a combination of shrinkage and gas pores was detected at the crack origin (Figure 7a). The crack-initiating pores and pore networks have a large projected area perpendicular to loading direction and a small distance to the specimen surface. The latter observation may be caused by higher stress intensities of pores near to the specimen surface compared to pores within the material [9].

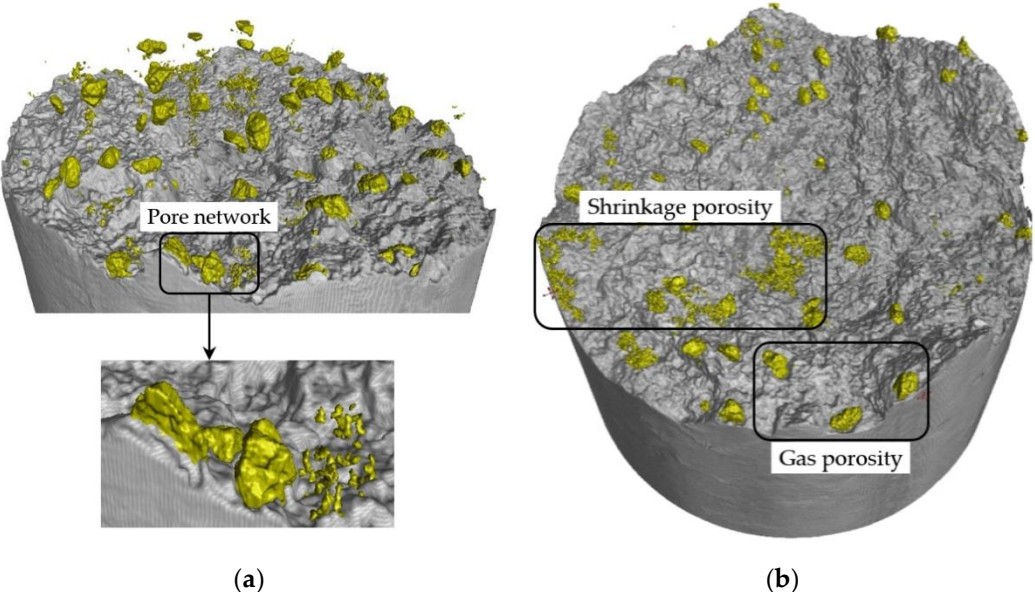

**Figure 7.** Fracture surface scanned by μCT: (**a**) Discontiguous crack-initiating pores (crack origin detected by SEM, specimen No. 09); (**b**) Exclusively pores on the fracture surface. (specimen No. 01).

Additionally, the influence of inner pores on the crack initiation behavior was evaluated by analyzing the pores on the fracture surface (Section 2.3). A combination of shrinkage and gas pores is present on the fracture surface (Figure 7b). Shrinkage pores are assumed to be more crucial for crack initiation due to their lower sphericity and larger projected area perpendicular to the loading direction.

The above observations indicate that extensive single pores and pore networks caused by the low cooling rate of the lost foam casting process are crucial for crack initiation. It is concluded that fatigue damage is the predominant failure mode compared to creep damage as large pores cause high stress intensities under cyclic loading, intensified by their interaction.

### 3.3. Pore Analysis on a Global Scale

Figure 8 shows the pore volume fraction $\phi_{Vol}$ and the number of pores $n_{pores}$ with a maximum Feret diameter $d_{Feret,max} > 100$ μm over the lifetime (number of cycles to a stress drop of 20%). The maximum Feret diameter $d_{Feret,max}$ is equal to the diameter of the outer sphere of the pore. There is a trend towards lower lifetimes for higher pore volume fractions, as consistently observed under each test condition. The second batch shows a higher average pore volume fraction (1.17%) than the first batch (0.99%). The number of pores within the scanned volume does not correlate with the lifetime. The second batch shows a lower average number of pores (1489) than the first batch (2150). Thus, the second batch contains fewer but larger defects caused by the casting process of the cylinder heads.

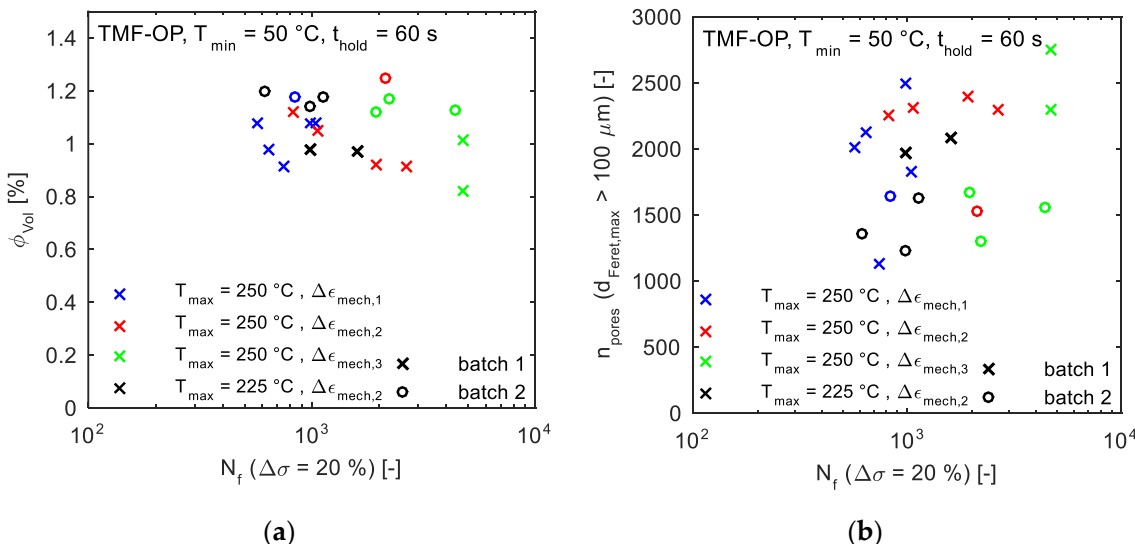

**Figure 8.** Global pore parameters vs. lifetime: (**a**) Pore volume fraction; (**b**) Number of pores with a maximum Feret diameter larger than 100 μm.

### 3.4. Pore Analysis on a Local Scale

Two characteristic parameters of single pores are considered: the maximum Feret diameter of the pore $d_{Feret,max}$ (Section 3.3) and the projected area of the pore to the plane perpendicular to loading direction $A_{proj}$ (Figure 9). Both parameters were derived from the μCT scans. It should be noted that by using $d_{Feret,max}$ and $A_{proj}$, the shape and orientation of the pore are not taken into account. Though the evaluation software provides the sphericity of the pore, there is no available information about the orientation of the pore.

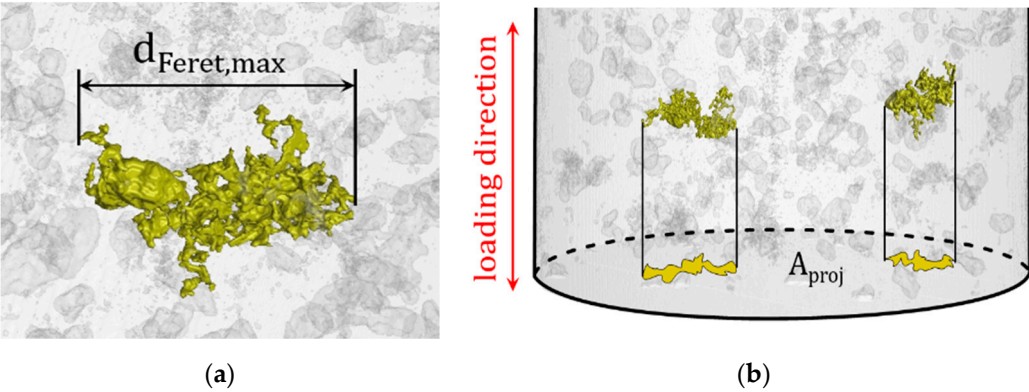

**Figure 9.** Characteristic parameters of single pores: (**a**) $d_{Feret,max}$; (**b**) $A_{proj}$.

For the statistical evaluation of the characteristic parameters, the exponential distribution is used [14] because of the presence of many small pores and less and less larger pores regarding the size of $d_{Feret,max}$ and $A_{proj}$. The exponential distribution is well suited to rank the actual crack-initiating pores detected by SEM as well as the pores on the fracture surface in the totality of pores within the volume. The equation for the exponential distribution is as follows:

$$d_{Feret,max,i} = -\ln(1 - P_i) \quad \text{and} \quad A_{proj,i} = -\ln(1 - P_i) \tag{1}$$

$P_i$ is the probability that the characteristic parameters are lower than the value of the corresponding $d_{Feret,max,i}$ and $A_{proj,i}$.

In Figure 10, $d_{Feret,max}$ and $A_{proj}$ are shown in the probability plot of the exponential distribution for specimens which were TMF tested at 1.2 $\Delta\varepsilon_{mech,2}$ and $T_{max} = 250\,°C$. For both $d_{Feret,max}$ and $A_{proj}$, the data lie approximately on a straight line, thus demonstrating the suitability of the exponential distribution. The data of the four specimens in Figure 10 are hardly discernable. It is concluded that there is no correlation between the course of the probability distribution of $d_{Feret,max}$ and $A_{proj}$ with the lifetime. Furthermore, the crack-initiating pores detected by SEM are marked with crosses. Notably, the actual crack-initiating pores of each specimen are not the pores with the largest $d_{Feret,max}$ and $A_{proj}$.

Additionally, Figure 11 depicts the characteristic parameter $A_{proj}$ of pores located on the fracture surface (Section 2.3) in the probability plot of the exponential distribution. For several specimens the pores with the largest $A_{proj}$ within the entire volume lie on the fracture surface (e.g., specimen No. 01, Figure 11a). On the other hand, there are some specimens where pores with the largest $A_{proj}$ within the entire volume do not lie on the fracture surface (e.g., specimen No. 07, Figure 11b). The same observations are made for the characteristic parameter $d_{Feret,max}$.

These investigations show that a proper criterion for crack initiation cannot be identified by exclusively studying single pores. There are two explanations for this observation: firstly, pores are present as pore networks, coalesce during a few cycles, and therefore act as crack initiator (see Section 3.2); secondly, shrinkage pores have discontiguous structures and therefore cannot be detected as one single pore by µCT. For these reasons, the sum of the projected areas $A_{proj}$ of discontiguous single pores at the crack origins detected by SEM and µCT (Figure 7a) is used as the projected area $A_{proj,sum}$ of the crack origin. All available projected areas $A_{proj,sum}$ are plotted in the probability plot of the logit distribution (Figure 12) [17]. The equation for the logit distribution is:

$$A_{proj,sum,i} = \frac{\sqrt{3}}{\pi} \cdot \ln\left(\frac{P_i}{1 - P_i}\right) \tag{2}$$

Since almost all data points lie within the 90% confidence region, the application of pore accumulations is considered appropriate.

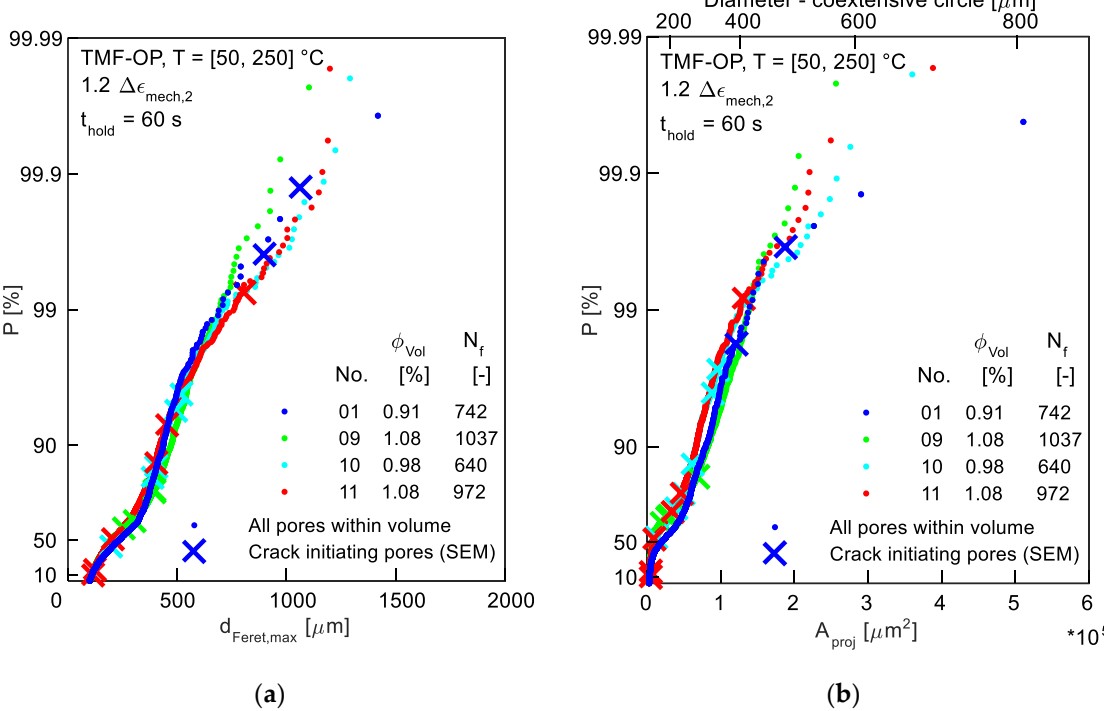

**Figure 10.** Characteristic parameters in probability plot of exponential distribution for specimens which were TMF tested at 1.2 $\Delta\epsilon_{mech,2}$ and $T_{max} = 250\,^{\circ}C$—crack-initiating pores detected by SEM are shown by the crosses. (**a**) $d_{Feret,max}$; (**b**) $A_{proj}$.

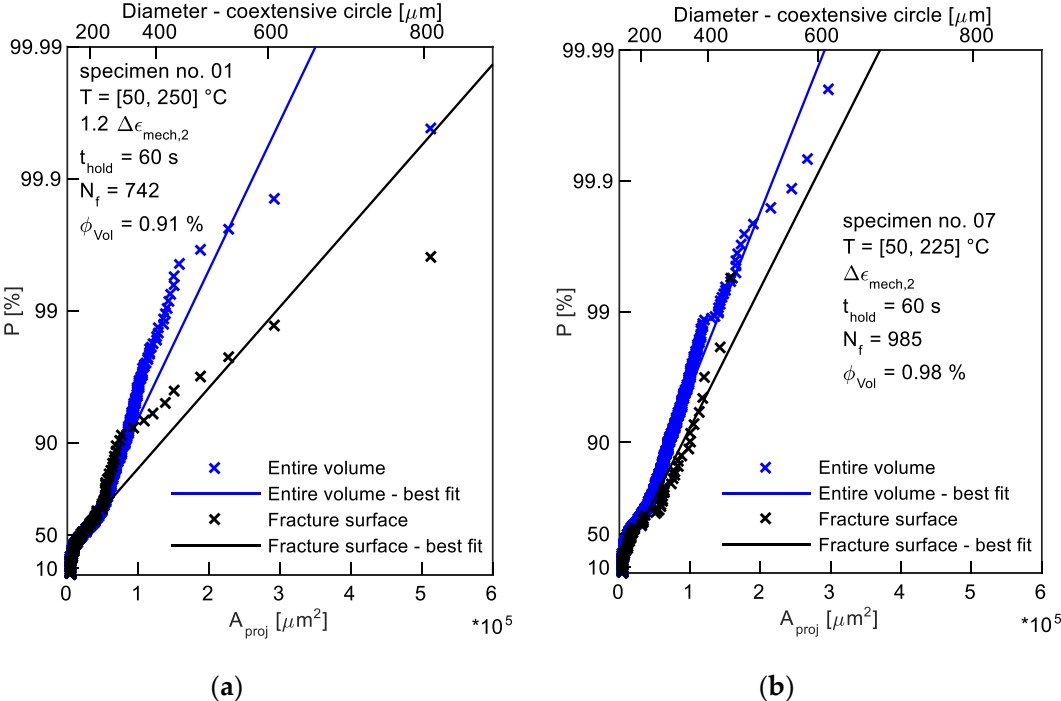

**Figure 11.** $A_{proj}$ in probability plot of exponential distribution—pores within the entire volume (blue) and exclusively on the fracture surface (black). (**a**) Specimen No. 01; (**b**) Specimen No. 07.

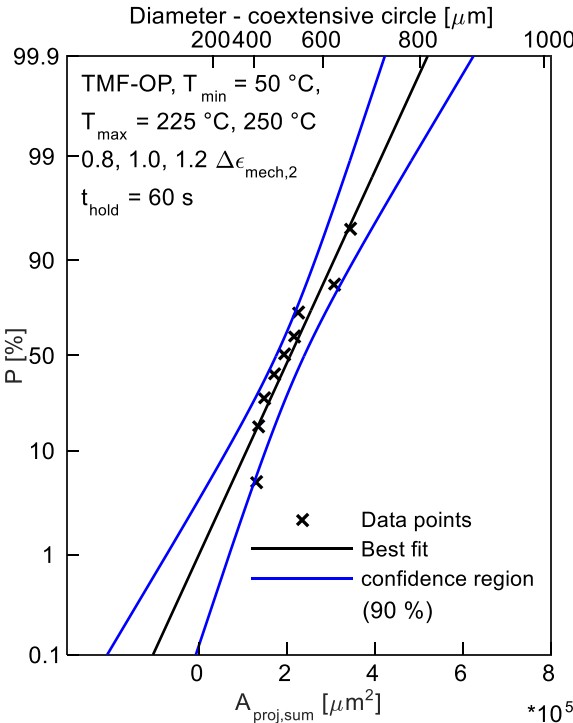

**Figure 12.** Sum of $A_{proj}$ of discontiguous single pores at crack origins detected by SEM and μCT in probability plot of logit distribution for all available specimens.

## 4. Computation of Pore Accumulations

### 4.1. Method of Computation

In order to assess the actual crack origin, a tool for the computation of pore accumulations was developed with the software MATLAB (software version R2015a, The MathWorks GmbH, Ismaning, Germany). For each pore, all surrounding pores within a defined inspection volume were detected. The inspection volume was defined in cylinder coordinates by $r_{iv}$ in radial direction, $\varphi_{iv}$ in angular direction, and $z_{iv}$ in axial direction (Figure 13b). In order to reach the best crack origin assessment by the computation of pore accumulations, different sizes of the inspection volume were manually tested (0.25 mm to 1 mm in radial direction, 10° to 30° in angular direction and 0.1 mm to 1 mm in axial direction). The tested sizes of the inspection volume were derived from the size of the crack origins detected by SEM. Additionally, the SEM investigations of the fracture surfaces showed that in all cases, crack initiation occurred near the specimen surface (see Section 3.2). Therefore, only inspection volumes touching the specimen surface were considered.

The projected areas $A_{proj}$ of the collected pores were summed up to $A_{proj,acc}$ for each inspection volume. Subsequently, the pore accumulations were sorted in descending order. The pore accumulation with the largest computed $A_{proj,acc}$ is marked green. The crack-initiating pore accumulation is marked red. All crack-initiating pores detected by SEM lie within the inspection volume of the corresponding pore accumulation. The projected areas are presented as area equivalent circles (Figure 13).

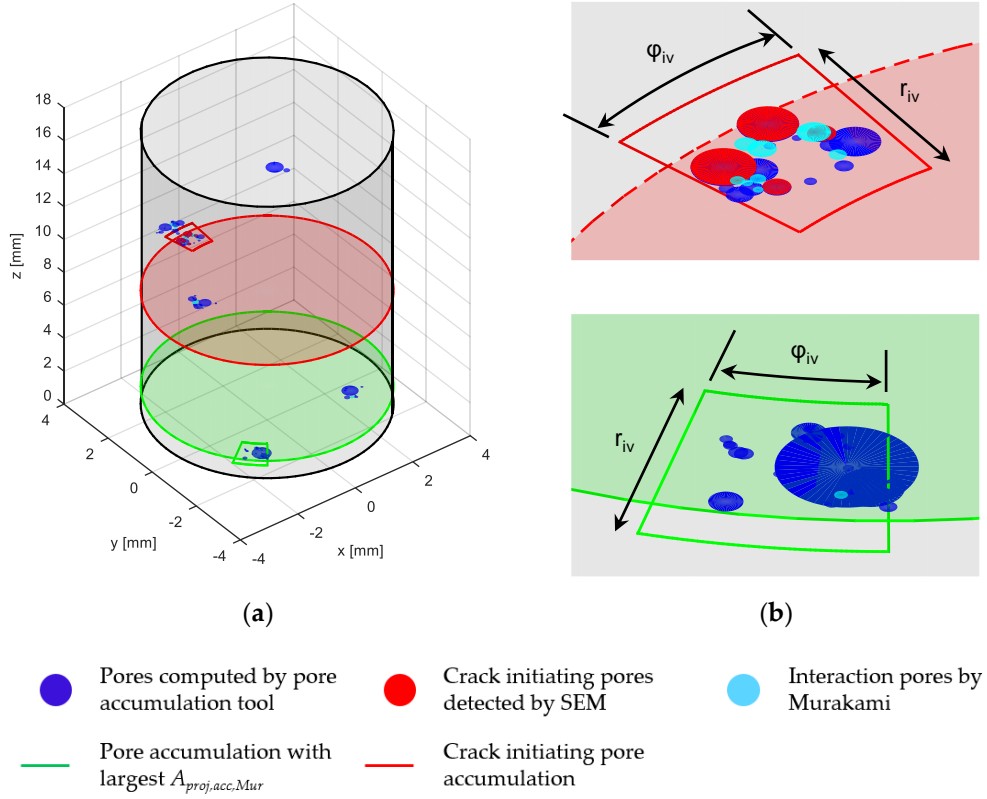

**Figure 13.** (**a**) Computed pore accumulations in terms of $A_{proj}$ for specimen No. 04; (**b**) Detail view of the crack-initiating pore accumulation including crack-initiating pores detected by SEM (top) and the pore accumulation with the largest computed $A_{proj,acc,Mur}$ (bottom).

Additionally, the described procedure is expanded using interaction criteria by Murakami [9]: for two pores having a smaller distance between each other than the diameter of the smaller one an additional pore is generated between the two pores (Figure 13). This method is based on the theory that the section between the pores can be interpreted as initial crack because two adjacent pores coalesce during a few cycles.

It should be noted that the provided method is based on the projected areas of the pores. Therefore, different pore shapes and orientations with their resulting higher stress intensities are not considered. Using the developed tool, spherical pores are evaluated as more and flat pores less dangerous than they are in reality.

### 4.2. Results of Computation of Pore Accumulations

In order to provide a measure for the suitability of computing pore accumulations with respect to assessing the actual crack origin, the computed pore accumulations are sorted in descending order according to the size of their accumulated projected areas $A_{proj,acc}$. From this ranking, the placement of the actual crack-initiating pore accumulation detected by SEM is determined (Table 1, column "$A_{proj,acc}$"). On the other hand, the single pores are sorted in descending order according to the size of their projected areas $A_{proj}$. From this ranking, the placement of the largest actual crack-initiating pore detected by SEM is determined (Table 1, right column). Comparing the computation of pore accumulations with the exclusive consideration of single pores, a smaller number indicates a better placement and thus a better predictive accuracy of crack origins. The approach with the best placement is marked grey for each specimen. Number '1' means that the actual crack origin is exactly matched. (Table 1).

For most specimens, the computation of pore accumulations is suitable for crack origin assessment because it improves the predictive accuracy compared to the exclusive consideration of single pores.

However, for two specimens (No. 07 and 08) the tool failed. There is no significant improvement of the results by the consideration of interaction pores by Murakami (Table 1, column "$A_{proj,acc,Mur}$"). A possible reason for that is the much larger projected area of the accumulated pores compared to the area between the pores. The inspection volume providing the optimal results has the dimensions $r_{iv}$ = 1 mm, $\varphi_{iv}$ = 15°, $z_{iv}$ = 0.5 mm.

**Table 1.** Evaluation of the suitability of computing pore accumulations for crack origin assessment (a smaller number indicates a better placement and thus a better predictive accuracy of crack origins; number '1' means that the actual crack origin is exactly matched).

| Specimen No. | Pore Accumulations | | | Single Pores |
|---|---|---|---|---|
| | Placement of Crack-Initiating Pore Accumulation in the Descending Order of the Computed Pore Accumulations | | | Placement of Largest Crack-Initiating Pore Detected by SEM in the Descending Order of Single Pores |
| | $A_{proj,acc}$ | $A_{proj,acc,Mur}$ | | |
| 01 | 1 | 1 | < | 2 |
| 02 | 6 | 5 | < | 6 |
| 04 | 3 | 3 | < | 79 |
| 06 | 7 | 9 | < | 37 |
| 07 | 32 | 35 | > | 8 |
| 08 | 2 | 4 | > | 1 |
| 09 | 26 | 12 | < | 149 |
| 10 | 4 | 4 | < | 25 |
| 11 | 4 | 6 | < | 6 |

## 5. Conclusions

The aim of this work was the identification of a criterion for crack initiation in order to provide the basis for a nondestructive quality control of TMF loaded porous components with improved statistical safety. TMF tests under realistic service conditions were performed on specimens extracted from automotive cylinder heads manufactured by lost foam casting. By performing μCT scans before and after the TMF tests, as well as comparing the fracture surfaces from SEM and μCT scans, the characteristics of crack-initiating pores were detected. Several characteristic pore parameters on the global and local scale were statistically examined in order to identify their influence on lifetime and crack initiation. Furthermore, a tool for computation of pore accumulations was developed. The following main conclusions can be drawn:

1. The positive mean stress in the TMF tests is caused by the out-of-phase loading condition and cyclic softening by ageing of the material.
2. Large shrinkage and gas pores, caused by the lost foam production process, are crucial for crack initiation.
3. Crack-initiating pores and pore networks have a small distance to the specimen surface and a large projected area to the plane perpendicular to loading direction.
4. On a global scale, the pore volume fraction and number of pores do not correlate with the lifetime.
5. On a local scale, statistical evaluation of single pores implied that pore accumulations are crucial for crack initiation.
6. For most specimens, the computation of pore accumulations is suitable for crack origin assessment because it improves the predictive accuracy compared to the exclusive consideration of single pores.
7. There is no further improvement of the predictive accuracy by consideration of interaction of pores by Murakami [9] because the projected area of the accumulated pores is much larger than the area between the pores.

A criterion for crack initiation in lost foam cast cylinder heads consisting of the application of pore accumulations under consideration of pore interaction was found. Applying the developed tool for the computation of pore accumulations to μCT scans of lost foam cast components enables the early-stage assessment of critical sections within the material. In industrial applications, the statistical power can be improved because of time-saving and cost-effective nondestructive testing (application of μCT) instead of time-consuming and cost-intensive destructive testing (application of TMF tests).

Even though, for most specimens, the computation of pore accumulations provides a better predictive accuracy of crack origins compared to the consideration of single pores, it can still be improved. A possible way is expanding the pore accumulation tool with the consideration of the edge distance of pores [10] since pores near to the specimen surface have higher stress intensities than pores within the material [9]. Furthermore, even though SEM investigations showed that crack initiation occurred near the specimen surface, it is planned to extend the applied inspection volume to the entire cross-section of the specimen to consider the possible influence of inner pores on crack initiation. Subsequently, a correlation between the final criterion and the lifetime can be derived in order to assess the lifetime variation of cylinder heads as a function of the variation of the criterion.

**Author Contributions:** Conceptualization, M.W., M.H., M.R. and H.-J.C.; formal analysis, M.W.; investigation, M.W.; methodology, M.W. and M.E.; project administration, M.W.; resources, A.M., M.E., M.H. and M.R.; software, M.W.; supervision, A.M., M.H., M.R. and H.-J.C.; validation, M.W., M.H., M.R. and H.-J.C.; visualization, M.W.; writing—original draft, M.W.; writing—review & editing, A.M., M.E., M.H., M.R. and H.-J.C.

**Acknowledgments:** The authors thank BMW AG for providing series cylinder heads as well as Manfred Hück, Stefan Slaby and Fabian Sartorius for their general support and discussions.

**Conflicts of Interest:** The authors declare no conflict of interest.

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
