# Peer review of "Thermomechanical Fatigue of Lost Foam Cast Al–Si Cylinder Heads—Assessment of Crack Origin Based on the Evaluation of Pore Distribution"

_metals, doi:10.3390/met9080821_

Reviewer 1 Report

Some parts of the Experimental results could be improved in order to gain in clarity. Please find my comments below.

Line 58: « studies [15,16] underlie » → studies [15,16] underline

Line 108: Please explain what do you mean by « freely solidified surfaces »?

Line 146: How is the crack origin identified? I understand that comparison of pores and fracture surface allows identifying which pores belong to the fracture surface but how are the pore(s) responsible for crack initiation identified?

Line 163: The trend « towards lower lifetimes for higher pore volume fractions » is mainly visible for batch 1. Do you have an explanation why the second batch is not so sensitive to the pores volume fraction?

Fig. 9a: the last number on the X axis seem to be cut.

Fig. 9b, Fig. 10: « × 10 » ? A number must be missing

Line 216: How is the size of the « inspection volume » defined? On the next page, it seems you test different sizes. What is the criterion retained to assess if the inspection volume is good or not?

Line 235 to 245: This paragraph and Table 1 are very difficult to understand. Please give more explanation.

I finally understood (after several reads) that first, pores accumulations or clusters are classified according to their size Aproj,sum. Then, the rank of the cluster of pores that was identified at the crack initiation site is reported in Table 1. If the rank is 1 then the method is successful and if it less than 12, it is still assumed « suitable ».

How do you choose the limit between suitable and unsuitable?

Please enhance the presentation of the different columns of table 1. I guess the 3rd column considers the rank of the crack initiating pore among the size classification for single pores and that for specimens 7 and 8, consideration of a single pore is better than consideration of clusters.

Except for specimens 1 and 8, the largest cluster (A proj,acc or A proj,acc,Mur) or the largest single pore never match with the crack origin. Do you have a proposition to enhance the prediction of the crack initiation site? In the conclusion you propose to take into account the « edge distance of pores ». Do you have some results that could sustain this proposition?

Reviewer 2 Report

A comprehensive empirical study was undertaken. Perhaps the main finding was, 'Crack initiating pores and pore networks have a small distance to the specimen surface and a  large projected area to the plane perpendicular to loading direction.' (261). While this is an intuitive result, it was good to see it explored empirically. Thank you for an interesting paper. My questions, in no particular order, follow. The material is apparently an ageing aluminium. Hence I assume a precipitation hardening alloy - this is confirmed at 79. Material needs to be named, or do you have a good reason not to provide this detail?   The possibility of precipitation in the microstructure has not been considered. Yes, I understand that your analysis is at the macroscale rather than the micro or grain level. Do you have any comment to make on the potential contribution of precipitates? I am not convinced the grounds are strong to attribute softening to ageing (257), at least not precipitation hardening. Please consider this wording carefully to avoid confusion with alloy ageing. Creep-fatigue has not been explicitly considered. Relative to the fatigue-creep spectrum, where do you believe the failures are positioned? Or to ask the question another way, what contribution do you believe the maximum temperature (line 97) made to the failure mechanisms? After all, we are talking about thermomechanical failure here. It would appear from the images that there is a high pore volume. I was not expecting so many defects. Have you deliberately selected a worst case scenario, or adjusted the process parameters to yield high porosity? Process parameters are important to note, where possible. I would have like to see summarised somewhere, a critique of what μCT, as used as a quality tool, can and can not detect. What are the limitations with the method? After all, quality was the rationale for the study. 113: ' Both scans were virtually overlapped to identify possible crack initiating pores. ' Was there any evidence of plastic deformation and how did you correct for that? Did you  ignore any deformation less than 50 μm (118), and if so what were the possible limitations of doing this? How confident are you of these matches? Looking at figure 3, which I presume is intended to show this overlap, I do not see any corresponding features, so I am  left doubting the robustness of this. What other failure mechanisms were at work? You wrote ' In many specimens a pore network was identified at the crack origin ' (146) but you do not state how prevalent this was or what the other failure origins were. How did you differentiate between shrinkage and gas porosity (157)? The projected area measurement (172) is a worst-case measure of the pore, since it does not consider the structure. A spherical spongy pore, and a flat disk void could have exactly the same diameters and projected areas. We would not expect them to behave in the same way under loading, yet your analysis considers them identical. I am not expecting the analysis to be redone, but I would expect that you would comment on how this might affect your results. Please explain/justify the use of the exponential (175). I don't believe that Figure 9 really shows 'perfectly' (182) straight lines. I don't believe they are good fits at all. 'statistics of extremes ' (281) Are you referring to the use of the exponential to model pore characteristics? Or to model life? 'in all cases crack initiation occurred near the specimen surface 219' Do you have an opinion as to why? Your results appear to show that porosity deep within the bulk is inconsequential? Do you feel that is a reliable interpretation? Given that the stress is uniform across the test samples, why are pores under the surface at greater risk than those more deeply buried? 184: 'course'? You claim to have developed an algorithm, 'an algorithm for computation of pore 255 accumulations was developed '.Yet you have not published it, and therefore it cannot be reviewed, and therefore cannot be an academic claim. 'a microstructural criterion 272 for crack initiation was found. ' What is this criterion? 'Applying this tool to μCT scans of Lost Foam cast components enables 273 the early-stage assessment of critical sections within the material by cost-effective, non-destructive 274 testing. ' I did not see any evidence of cost effectiveness, so I suggest this claim be dropped.

Reviewer 3 Report

The paper deals with impact of distribution of defects in a typical lost foam casting Al-alloys. The paper is of a very good quality for both content and writing. The method consists in combining TMF testing and µCT analysis before and after cycling to address a combined statistical and damage analysis. Among very nice results developed by the authors it is worth noting that they gave evidence that the crack initiating pores are not the pores with the largest Feret diameter. An alternative procedure has been proposed taking into consideration pore accumulations instead of single pores. 
Even though these results should be more developed it appears as a very good way to predict pores network influence on TMF life. mandatory revision to be adressed before publication • loading cycle should be plot to clarify testing conditions • strain level should be precised to ascertain damage mechanisms, thus the assertion "However, there are higher lifetimes due to lower maximum stresses with 
increasing maximum temperature." is a bit speculative without clear results on strain-stress loops by comparing the different conditions tested. What is about the strain energy density? • references to work in the very narrow field of the paper are missing :  o it is stated that "the above observations indicate that porosities of large size caused by the low cooling rate of the Lost Foam casting process are crucial for crack initiation. " which point has been clearly discussed in S. Dezecot, et al, 3d characterization and modeling of low cycle fatigue damage mechanisms at high temperature in a cast aluminum alloy, Acta Materialia 123 (2017) 24–34. 
 o besides, it is stated that "it is common to use statistical methods to evaluate pore distributions because of their suitability 
for volume extrapolation [...] and extreme values [analysis]" that have been thorougly  
documented in . Wilson, et al, Isothermal fatigue damage mechanisms at ambient and elevated temperature of a cast al-si-cu aluminium alloy, International Journal of Fatigue 121 (2019) 112–123. 
 o reference to method of defects and image analysis, including Feret diameter, should be added
